# Toward a Better Understanding of Phosphorus Nonpoint Source Pollution from Soil to Water and the Application of Amendment Materials: Research Trends

**Xiaofei Ge [1], Xingyu Chen [1], Mingxin Liu [1], Chensi Wang [1], Yingyu Zhang [1], Yukai Wang [1], Huu-Tuan Tran [2], Stephen Joseph [3] and Tao Zhang [1,*]**

[1]   Beijing Key Laboratory of Farmland Soil Pollution Prevention and Remediation, Key Laboratory of Plant-Soil Interactions of Ministry of Education, College of Resources and Environmental Sciences, China Agricultural University, Beijing 100193, China

[2]   Department of Civil, Environmental and Architectural Engineering, University of Kansas, Lawrence, KS 66045, USA

[3]   School of Materials Science and Engineering, University of New South Wales, Sydney, NSW 2052, Australia

*   Correspondence: taozhang@cau.edu.cn; Tel.: +86-10-6273-3638

**Abstract:** Phosphorus (P) nonpoint source pollution from soil to water is increasing dramatically, leading to the eutrophication of water bodies. Using amendment materials for P retention in soil is a promising strategy for environmental restoration and nonpoint source pollution management. This strategy has attracted significant attention because of its highly effective P retention. This study reviews management strategies of P nonpoint pollution from soil to water, including the basic P forms and accumulation situation in soil and P loss from soil to water. Recent advances in the use of amendment materials, such as inorganic, organic, and composite amendment materials, to mitigate P pollution from soil to water have also been summarized. Environmental risks of reloss of P retention in soil with different soil properties and water conditions have also been investigated. This review improves the understanding of P nonpoint source pollution from soil to water, providing an innovative perspective for the large-scale application of amendment materials to control water eutrophication.

**Keywords:** phosphorus; nonpoint source pollution; amendment materials; water

## 1. Introduction

In an era of rapid economic and technological development, resource scarcity has become an inevitable problem for global sustainable development [1–4]. Phosphorus (P) is an indispensable element for organisms to conduct their life activities [5–9]. The use of P fertilizer can supplement the effective P in soil, increase crop yield and quality, and maintain food security and sustainable development [10–12]. The dependence of modern agriculture on inorganic phosphate fertilizer will inevitably increase the demand for nonrenewable phosphate rock [13,14]. According to the data from the United States Geological Survey in 2022 (Figure 1), the global base reserves of phosphate rock are 69 billion tons. China's current phosphate rock reserves are 3.2 billion tons, accounting for less than 5% of the global total reservation amounts, and approximately 70% are recognized as refractory phosphate rock. As the country with the largest consumption of inorganic fertilizers in the world, China uses 80–90% of the mined phosphate ore as mineral phosphate fertilizers in food production, and crops absorb only approximately 20% of this P amount, which eventually enters the food consumed by people [15]. Meanwhile, the high demand for inorganic fertilizers, along with low nutrient utilization efficiency [16], has put significant pressure on the limited P rock reserves and the inorganic fertilizer market that depends on these reserves.

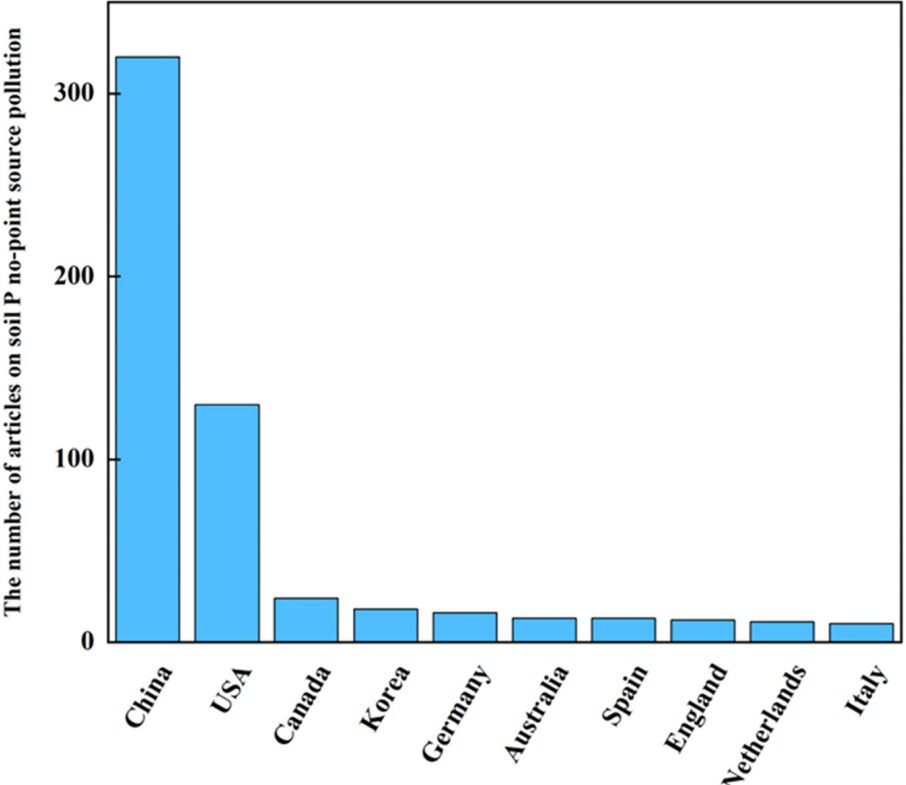

**Figure 1.** The number of articles on P nonpoint source pollution from soil to water.

In addition, due to the wide range and prevalence of agricultural activities, a large amount of P accumulation results in farmland nonpoint source pollution, which is considered one of the main sources of water environmental pollution risks [17,18]. Phosphate fertilizer applied is easily absorbed by soil particles or forms precipitates with calcium, magnesium, iron, and aluminum plasma in soil. However, the P absorption capacity of soil with excessive P is close to saturation, which increases unstable P content in the soil and, thus, increases the risk of P loss to water (Figure 2) [1]. The transfer of P from the farmland to the water environment results in severe agricultural nonpoint source pollution [19]; it can dominate the eutrophication of aquatic ecosystems [20].

For a long time, the P load of freshwater ecosystems has been increasing with the process of urbanization and industrialization. A large amount of nitrogen, P, and other nutrients enter water bodies, causing serious environmental pollution problems and leading to water degradation and reduction of aquatic biodiversity. The eutrophication degree of water bodies is improved, which provides nutrients for algae and other plankton, especially cyanobacteria. A large number of algae will reduce the oxygen content of water bodies and release algal toxins into water bodies, threatening other aquatic organisms. Eutrophication and harmful algal blooms (HAB) are widely considered two of the biggest water quality problems at present. Eutrophication can be divided into natural eutrophication and artificial eutrophication. Under natural conditions, lake sediments are constantly increasing, leading to nutrient accumulation and eutrophication. This process usually takes hundreds or even thousands of years. Therefore, an important measure to reduce water eutrophication is to reduce anthropogenic nitrogen and P emissions.

A large amount of applied P fertilizer is directly lost into river water bodies without plant absorption, leading to a series of water ecological pollution problems, such as the destruction of water ecosystem biodiversity and serious eutrophication of water bodies. The main reason is the management omission of P fertilizer application in agricultural production. The efficiency of P fertilizer utilization in many rural areas is generally low. River nonpoint source TP pollution area is widely distributed and is mainly concentrated

in the agricultural production and living areas where soil erosion is more serious. After soil erosion, the massive input of chemical fertilizers and pesticides and poor management, as well as rural livestock and poultry-breeding discharges, have made the problem of non-point source TP pollution in river water bodies increasingly serious, and nonpoint source pollution has become one of the most important causes of river water quality deterioration.

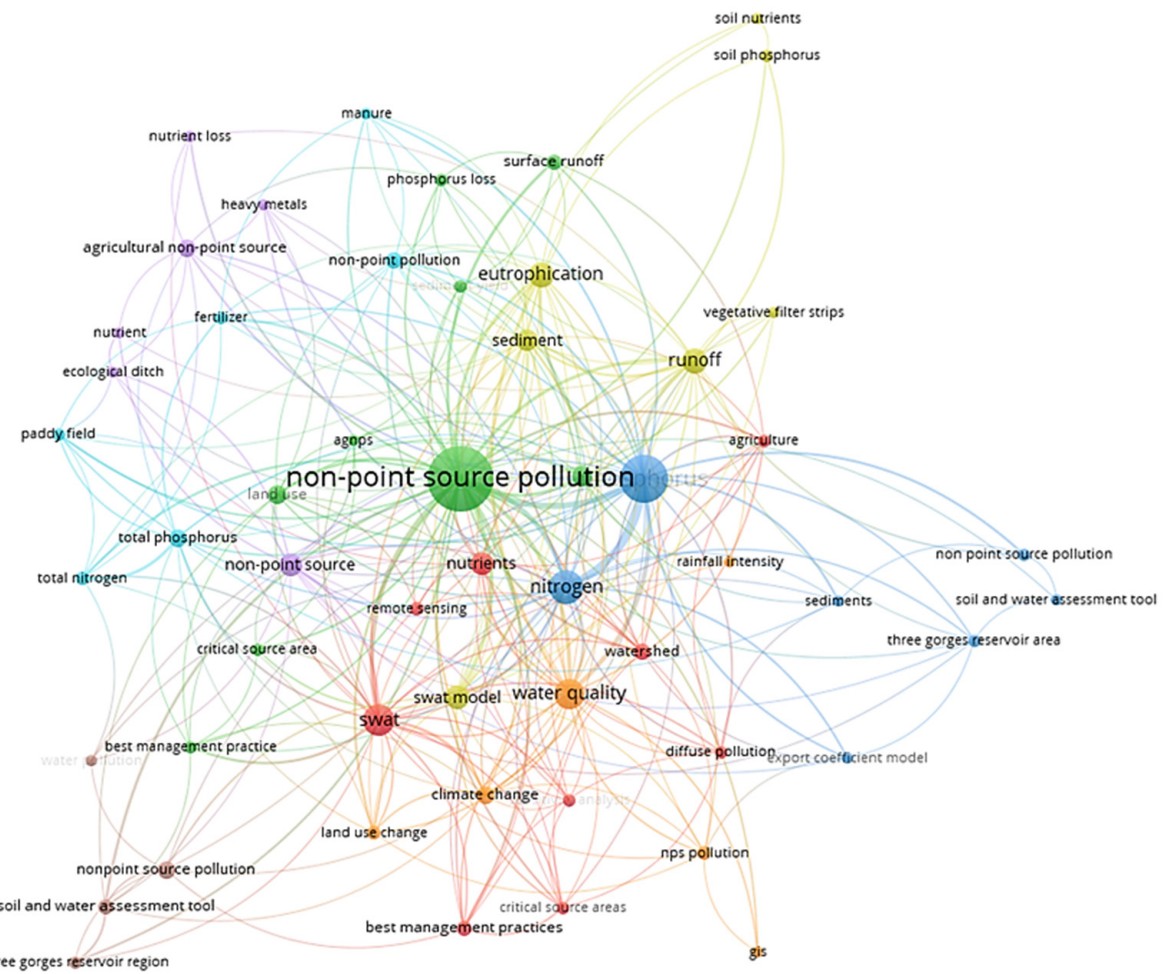

**Figure 2.** Hop map of keywords for articles on P nonpoint source pollution from soil to water using VOSviewer1.6.18 software.

China is a largely agricultural country with a fertilization amount of 100 million tons. The average fertilization amount is 2.6 times the world average, but the fertilization utilization rate is only 30–50% of the world average. A great amount of phosphorus enters water bodies through various channels. Usually, the contribution of P in water bodies from various external sources in descending order is agricultural drainage, domestic sewage, and industrial sources of pollution [21].

Numerous studies have shown that nutrient losses from agricultural production are the main source of nutrient surpluses in agricultural ecosystems, water bodies and wetland ecosystems, and, therefore, the largest cause of phosphorus pollution in water. TP emissions from agricultural sources amounted to 284,700 t, accounting for 67.27% of the total national emissions. According to the latest data from the Food and Agriculture Organization of the United Nations (FAO), China was the country that applied the most nitrogen and P fertilizers in the world during 2002–2017, exceeding the sum of the fertilizer applied by the second- and third-ranked countries. China's arable land area only accounts for about 7% of global arable land, but the amount of P fertilizers applied to agriculture accounted for 30.09% of the world's fertilizer utilization (2002). It is noteworthy that the in-season

utilization rate of fertilizers in Chinese agricultural production is low, with the in-season utilization rate of P fertilizer ($P_2O_5$) being around 20%. This means that some of the nitrogen and P nutrients that are not absorbed and used by crops and leave the cultivated soil will be further lost to surface water, groundwater, and the atmosphere through surface runoff, leaching, ammonification, nitrification, and denitrification. This will cause environmental impacts such as nitrate pollution of groundwater and eutrophication of water bodies.

In agricultural ecosystems, fertilizer and animal manure are the main sources of P input, so the two main sources of P emissions from agricultural sources can be judged to be excessive fertilizer application and animal manure. In terms of source composition, based on the national multiyear average, the proportion of P emissions from agricultural sources is 45.73% from overfertilization and 54.27% from livestock and poultry manure, indicating that the primary cause of agricultural surface pollution in China is livestock and poultry manure emissions.

When overfertilization or manure overload occurs in farmland (or grassland) ecosystems, nitrogen and P in different compositional forms enter surface and groundwater mainly through surface runoff and subsurface leaching, which, in turn, leads to surface pollution such as eutrophication of water bodies [22].

Farmland nonpoint source pollution affects 30–50% of the global land area [1]. In the United States, nonpoint pollution accounts for two-thirds of the total pollution [23]. Fertilizer and manure inputs have exceeded crop removal rates by up to 50% in many agricultural regions [24]. The input of TP from agricultural nonpoint sources to the North Sea in Europe accounts for 25% of the flux to the sea. Among various sources, TP from agricultural nonpoint sources in the Netherlands accounts for 40–50% of water pollution. P loads introduced by nonpoint sources in 270 rivers in Denmark accounted for 52% of total pollution [25]. Therefore, an important way to control P nonpoint source pollution from soil to water is to control P loss in soil.

Recently, many scholars have researched P nonpoint source pollution from soil to water and the application of amendment materials. We selected "soil P nonpoint source pollution" as the keyword for screening on the Web of Science website, and selected the literature data from 2013 to 2023 to analyze with the country region; the results are shown in Figure 1. China has published approximately 56.4% of the publications on soil P nonpoint source pollution. As shown in Figure 2, the keyword "P nonpoint source pollution" in articles on the Web of Science database is strongly related to the keywords "nitrogen" and "water quality." This explains the consistency of the global agricultural nonpoint pollution situation with the research hotspots described in the Web of Science database. This shows that the management of P nonpoint source pollution from soil to water is receiving increasing attention in China.

Research has been conducted on applying environment-friendly amendment materials that can absorb large amounts of P to convert unstable P into medium-stable P in soil [26–28]. Adding iron, aluminum, and calcium to P improves soil P retention ability and reduces soil P loss to water [29]. This effective and environmentally sound treatment method that can avoid the ecological risk of P loss. In this study, P pollution regulation from soil to water using specific amendments has been discussed and summarized. This review provides critical information about P pollution regulation and amendment material application, which is useful for agricultural nonpoint source pollution management.

## 2. Phosphorus Nonpoint Source Pollution Management Strategies

### 2.1. Phosphorus Forms and Accumulation in Soil

P exists in two forms in soil: inorganic P and organic P. Most P in arable soils is mainly inorganic P, which plants can absorb and use [30]. Organic P can be converted to inorganic P after mineralization. Inorganic P exists in soil in three forms: orthophosphate, pyrophosphate, and polyphosphate. Among these three forms, orthophosphate is the main component. Soil P can also be divided into mineral P, adsorbed P, and water-soluble P, according to its binding characteristics with the soil matrix. Mineral P is the most difficult

P form to be absorbed and utilized by plants, mainly occurring in apatite. Adsorbed P is mainly adsorbed by soil clay, Fe-Al oxide, hydrated oxide, carbonate, or organic matter by physical actions in the form of $H_2PO_4^-$ and $HPO_4^{2-}$. Water-soluble P can be directly absorbed and utilized by plants, and its content is generally low, which is affected by the dissolution of mineral phosphorus and the release of adsorbed phosphorus. The difference in soil types frequently leads to different organic P proportions, which generally account for 20–80% of soil TP [31]. Soil organic P can be divided into combined organophosphorus, adsorbed organophosphorus, and microbial P, including phosphoric acid, phosphorus ester, stable nucleic acid, phosphoprotein, and microbial P. Among them, inositol phosphate is the main organic P form, accounting for approximately 50% of the total organic P.

*2.2. An Important Source of Phosphorus to Water: Soil Phosphorus Loss*

P loss from soil to water occurs in three ways: surface runoff, soil erosion, and leaching, all of which end up in the water and cause eutrophication. The amount of P loss is related to rainfall, rain intensity, soil texture, and P background value in soil. The results of the first survey of pollution sources in China show that from 2012 to 2014, the average total P loss in China was $1.08 \times 10^5$ t [32].

Algae are very sensitive to P in water, and a small amount of soil P in water ($0.05$–$0.10$ mg·L$^{-1}$) can lead to water eutrophication. P migration from soil to water includes water-soluble P and soil particle-combined P. Generally, the low solubility of soil P, the high adsorption capacity of clay for P, and the strong combination of P and soil organic matter make the water-soluble P content in the soil low. The P migration is low, and most soil P is in the form of particle-combined P. However, in soil containing excessive P, the absorption capacity of soil for P is close to saturation, and water-soluble P content increases [33]. Therefore, the risk of P loss from soil to water due to leaching increases. McDowell et al. [34] also concluded that leaching causes as much, and sometimes more, P loss from farmland soil to water as surface runoff and soil erosion.

In recent years, P loss from soil has seriously affected P levels in water bodies, and many scholars have paid increasing attention to P loss in arable land. Different indicators have been used to evaluate P loss from soil to water, mainly including direct monitoring and soil P index prediction [35,36]. Direct monitoring is based on simulated rainfall, laboratory soil column leaching, and long-term monitoring tests, which evaluate P loss from soil to water by measuring TP in runoff or groundwater and leaching solutions. The risk of P loss from soil to water is assessed by predicting and assessing the soil P index. There are two methods: single-index prediction and multi-index combination prediction.

For single indicator prediction, different soil test P (STP), such as $CaCl_2$-P, Olsen-P, Bray-P, and M3-P, is used to determine the P content in a specific part of the soil, and then various P contents are evaluated according to the established threshold value. Beyond the threshold value, the risk of P loss threatening water quality increases significantly. Based on the relationship between soil P content and water environment P, the piecewise linear model is generally adopted, and the discontinuity point is the threshold of P loss from soil to water. With an increase in the STP concentration, the P concentration in leachates increases slowly at first and then increases sharply when it exceeds the threshold value [37]. For example, the long-term Broad balk test showed that when soil Olsen-P content was less than 60 mg·kg$^{-1}$, the corresponding total P concentration in the soil drainage decreased ($<0.15$ mg·L$^{-1}$), but when soil Olsen-P content was greater than 60 mg·kg$^{-1}$, the total P concentration increased linearly. The threshold value of soil environmental P varies regionally, which is related to soil texture, planting methods, soil management methods, and other factors. In addition, soil $CaCl_2$-P is a capacity index that can be easily released into the water and can directly reflect the risk of P loss from soil to water. Therefore, the threshold of P loss from soil to water can also be determined on the basis of the linear relationship between other STP and $CaCl_2$-P.

Multi-index combination prediction relates the P loss risk to the intensity factor (the amount of soil P extracted using a particular method) and the capacity factor (the max-

imum amount of P that soil can absorb), i.e., the soil adsorption degree of phosphorus saturation (DPS), in which the intensity and capacity factors are measured in different ways. The capacity factor can be obtained from the P adsorption characteristic curve and the combination of weak crystal hydroxyl ferric oxide and hydroxyl alumina ($Fe_{ox}$, $Al_{ox}$), measured using the ammonium oxalate-oxalate solution extraction method. The intensity factors include M3-P, Olsen-P, Pox, etc. [38]. The threshold value of DPS is typically 25%, and the risk of P loss from soil to water increases rapidly when more significant than this value. In acidic soils, 12.5% is commonly used as the threshold of DPS [39]. In addition, some studies have used $Fe^{2+}$:P to predict the risk of P loss from soil to water, but it was mainly based on soil pore water [40].

*2.3. The Management Strategies for Soil Phosphorus Loss to Water Body*

In the 1980s, China identified the problem of soil P loss to water. There are several methods for reducing the risk of P loss from agricultural fields: reducing P inputs, reducing P loss to water, conservation tillage, and buffer zone construction. Increased and stabilized crop output is achieved through the yield response method, fertilization management, and nutrient balance method, reducing environmental problems caused by excessive fertilization [20]. Regarding P management, European Union researchers have proposed several strategies for achieving the goal of global sustainable P use, including adjusting P input, reducing P loss to water, increasing the cycle of P biological resources, recycling P products, and P transfer in the food chain.

Reducing P inputs includes reducing fertilization, precise fertilization, and limiting the use of organophosphorus pesticides. Soil organophosphorus loss is an important factor in water eutrophication and is caused by organophosphorus pesticides.

Under specific soil and runoff conditions, P is mainly lost through surface runoff. To a certain extent, preventing surface runoff can increase soil resistance to erosion and reduce the risk of soil P loss to water, including covering the soil surface with crop residues and setting buffer zones and grassland waterways [33]. However, these strategies effectively reduce particulate P transport but are ineffective for reducing dissolved P in high-P residual soil. In addition, these strategies take a long time for soil P concentrations to decrease, during which large amounts of dissolved P are lost [41].

The loss of P to water can be reduced by changing the form of P in soil. A quick and effective method is to add amendment materials to the soil to change the unstable P component in the soil, which is prone to P loss from soil to water, into a stable P component, thereby reducing the risk of P loss from soil to water.

## 3. Application of Amendment Materials for Phosphorus Nonpoint Source Pollution from Soil to Water

The transformation of P is in a dynamic equilibrium, including adsorption–desorption, precipitation–dissolution, and the action of soil microorganisms. Generally, the role of soil microorganisms is relatively weak, whereas chemical precipitation and physicochemical adsorption play a key role. For example, in acidic soils, iron, aluminum oxides, and hydroxides are the main substances for P retention. Clay minerals can immobilize P by the same mechanism. P retention in alkaline and calcareous soils is typically associated with the formation of calcium-bound P and can be fixed by $Al^{3+}$ or $A1(OH)_3$ in clay minerals. Inorganic matter, organic matter, and composite matter can be extracted from the waste and applied to the soil as modified materials to control P loss from soil to water. For example, the alumina could be extracted from alum sludge using different chemical treatment methods after being subjected to different thermal treatments [42]. Red mud is a byproduct of the Bayer process to extract alumina from bauxite [43]. Wheat straw biochar and corn straw biochar can be produced from wheat rice and corn straw, respectively. Fly ash is solid waste after the combustion of pulverized coal in coal-fired power plants [44]. These are modified materials converted from solid waste.

P amendment materials can accelerate the transformation of soil P from an unstable state to a medium or highly stable state, thereby reducing the risk of P loss from soil to water. According to their properties, amendment materials can be divided into three categories: (1) inorganic materials, including natural minerals, chemical materials, and industrial waste; (2) organic materials; and (3) composite materials. P retention is primarily affected by soil type, soil P background value, material properties, and material addition amount. Although amendment materials are effective in soil P retention based on laboratory-scale research, long-term P retention information about amendment materials at the farmland scale is lacking.

### 3.1. Inorganic Amendment Materials

P retention in the soil strongly correlates with extractable iron, aluminum, and $CaCO_3$ content in acidic soils [45]. The P retention mechanism of inorganic materials is primarily by increasing the concentration of Fe, Al, Ca, and Mg ions in soil and, thus, transforming the unstable P in the soil into medium and high stable P under the action of adsorption, precipitation, ligand exchange, and electrostatic attraction. In addition, some materials with good pore structure and significant specific electrostatic force and ion exchange properties, such as bentonite and zeolite, can effectively adsorb phosphate in soil, thereby reducing P loss. According to their composition, inorganic materials are classified into iron and aluminum materials, calcium and magnesium materials, clay minerals, and other materials.

#### 3.1.1. Calcium and Magnesium Inorganic Amendment Materials

Calcium and magnesium materials, including dolomite, desulfurized gypsum, gypsum, lime, calcite, and so on, can be applied for P retention from soil to water (Table 1). There are two mechanisms for P retention from soil to water when using calcium materials: (1) an increase in the $Ca^{2+}$ concentration in soil that can promote the formation of calcium phosphate precipitation, thereby reducing the solubility of P; and (2) increasing soil pH. Adding dolomite can promote the adsorption or precipitation of unstable P in soil on the surface of calcium ions and increase stable calcium and P compounds by increasing soil pH [46,47]. Adding desulphurized gypsum can increase the content of insoluble calcium phosphate ($Ca_8$-P and $Ca_{10}$-P) in soil, and then effectively control the dissolved P in the soil [47]. The amount of material added is an important factor affecting soil P retention. When the amount of lime is small, the release of calcium and magnesium ions is insufficient to promote the formation of phosphate precipitation, thereby increasing P content during leaching. For example, 2% addition (*w/w*) of slaked lime dust can reduce P loss by 77.2%, whereas 1.5% can increase P loss by 236% [46]. Calcium and magnesium can also be applied to soil to improve soil porosity and aggregate strength, thereby improving gas exchange and increasing crop yields.

**Table 1.** Different Ca/Mg inorganic amendment materials for P retention from soil to water.

| Ca/Mg Inorganic Materials | Soil Type | pH | Addition Amount (*w/w*) | Retention Situation | References |
|---|---|---|---|---|---|
| calcium carbonate | red soil | 5.42 | 2.0% | the soil Olsen-P contents increased by 33.9% | [48] |
| dolomite | red soil | 5.42 | 2.0% | the soil Olsen-P contents increased by 66.3% | [48] |
| dolomite | Calcareous soil | 7.56 | 2.0% | the soil $CaCl_2$-P content was reduced by 57.8% | [48] |
| dolomite | Calcareous soil | 7.6 | 5.0% | soil available P decreased by 3.10% | [49] |
| dolomite | Calcareous soil | 7.9 | 1.25% | the total dissolved P of leachate decreased by 68.4% | [50] |
| magnesia | Sandy | 7.1 | 2.0% | soluble P was reduced by 78.6% | [51] |

### 3.1.2. Iron and Aluminum Inorganic Amendment Materials

Iron and aluminum amendment materials for P retention from soil to water, include alum, red mud, ferrous sulfate, aluminum sulfate, fly ash, wastewater treatment residues, and so on (Table 2). Iron and P can form inner- and outer-sphere complexes through ligand exchange and electrostatic attraction to effectively regulate the transport of P in soil [51]. Red mud can fix phosphate on the existing surface by complexation, and then, through the adsorption of metal cations, such as $Al^{3+}$, $Fe^{3+}$, and $Ca^{2+}$, it generates metal oxide hydrate sites for further phosphate adsorption, eventually leading to surface precipitation or multilayer adsorption [52]. The establishment of multilayer adsorption depends on the availability of polyvalent metal ions during red mud leaching, mineral dissolution rate, surface species, solution pH, and total surface area [53]. Brennan et al. [52] found that in sandy, clay, and loam soils, adding 5-t·hm$^{-2}$ red mud could effectively reduce water-soluble P in soil, and the effect was the most significant in high-P soil. Aluminum is an element with a strong affinity for P, and the addition of 20-g·kg$^{-1}$ $KAl(SO_4)_2 \cdot 12H_2O$ can significantly reduce the content of unstable P in soil while increasing the content of moderately unstable P 30. At different soil pH values, the mechanism of P retention of aluminum-based materials differs. In calcareous soil, alum mainly forms poorly crystallized hydroxyl aluminum to adsorb unstable P, whereas, in acidic soil, it is fixed by the precipitation of $Al^{3+}$ and unstable P [54]. The main components of fly ash are silica, alumina, and iron oxide, among which rich iron and aluminum can increase the adsorption and precipitation of P in acidic soil. After the application of fly ash, inorganic P in soil is generally converted into Ca-P ($H_2SO_4$-P), NaOH-Pi, and residual P [26]. It can also reduce soil swelling and clay dispersion, as well as P loss to water, by reducing soil erosion.

**Table 2.** Different Fe/Al inorganic amendment materials for P retention from soil to water.

| Fe/Al Inorganic Materials | Soil Type | pH | Addition Amount (*w/w*) | Retention Situation | References |
|---|---|---|---|---|---|
| alum | Calcareous soil | 7.56 | 2.0% | the soil CaCl2-P content was reduced by 77.0% | [48] |
| alum | red soil | 6.04 | 2.0% | the soil CaCl2-P content was reduced by 93.8% | [48] |
| alum | | 7.6 | 5.0% | the soil CaCl2-P content was reduced by 91.9% | [49] |
| aluminum sulfate | red soil | 7.25 | 0.2% | the total dissolved P of leachate decreased by 80.6% | [55] |
| ferrous sulfate | red soil | 7.25 | 0.2% | the total dissolved P of leachate decreased by 80.6% | [55] |
| Al-WTR | peat soil | 3.8 | 10% | P adsorption maxima was increased by 11% | [56] |

### 3.1.3. Clay Mineral Inorganic Amendment Materials

P retention clay minerals include bentonite, zeolite, hydrotalcite, and so on (Table 3). The P retention mechanism by clay minerals is mainly by chemical adsorption. The higher the calcium, aluminum, and iron content in the elemental composition, the stronger the P adsorption capacity, but the phosphate integration performance is poor. As a result, these clays need to be modified physically and chemically to make them easier to absorb phosphate. Layered double hydroxide (LDH), also known as hydrotalcite compounds or anionic clay, is a promising P adsorption material [28]. It is composed of two-dimensional layered mixed hydroxides with the advantages of a permanent positive charge between layers, high anion exchange capacity, large specific surface area, and water resistance of the structure [57]. The mechanisms of phosphate adsorption include electrostatic attraction, ligand exchange, hydrogen bonding interaction, and ion exchange [58]. The direct application of LDHs through tillage on soils with a high-P application rate could fix P in soil through its strong adsorption capacity and reduce P transport to nearby water systems.

**Table 3.** Different clay mineral inorganic amendment materials for P retention from soil to water.

| Clay Mineral Inorganic Materials | Soil Type | pH | Addition Amount (*w/w*) | Retention Situation | References |
|---|---|---|---|---|---|
| Mg-Al LDHs | | 7.08 | 2% | the P effluent mass balance decreased 82.7% | [28] |
| natural zeolite | inceptisol | 6.4 | 5% | P was removed from the solution by 3.6% | [59] |
| lanthanum modified zeolite | aquatic soil | 7.9 | | the soluble reactive P in decreased by 86.9% | [60] |
| CFL-Z | aquatic soil | | | the P content in overlying water was reduced by 97.3% | [61] |
| montmorillonite | sandy clay loam | 7.8 | 1% | the calcium chloride-extractable P content was reduced by 62.8% | [62] |
| zeolite | sandy clay loam | 7.8 | 3% | the water-extractable P content was reduced by 9.9% | [62] |
| vermiculite | sandy clay loam | 7.8 | 3% | the Olsen-extractable P content was reduced by 79.8% | [62] |
| bentonite | Sandy loam soil | 8.28 | 10% | P maximum sorption capacity increased by 42.7% | [63] |
| kaolinite | Sandy loam soil | 8.28 | 10% | P maximum sorption capacity increased by 77.5% | [63] |
| zeolite | Sandy loam soil | 8.28 | 5% | P maximum sorption capacity increased by 70.0% | [63] |

### 3.1.4. Waste Inorganic Amendment Materials

Some inorganic waste materials can be used as P retention inorganic amendment materials (Table 4). Irshad et al. [64] evaluated the effects of waste inorganic amendment materials, coal ash, and wood ash on P retention from soil to water. It found that coal ash and wood ash can reduce water-soluble P in soil. Faridullah et al. [65] investigated the effects of waste inorganic amendment materials, wood ash, and sawdust, on P retention from soil to water.

**Table 4.** Different waste inorganic amendment materials for P retention from soil to water.

| Waste Inorganic Materials | Soil Type | pH | Addition Amount (*w/w*) | Retention Situation | References |
|---|---|---|---|---|---|
| fly ash | inceptisol | 6.4 | 5% | P was removed from the solution by 97.0% | [59] |
| coal ash | loamy sand | 7.6 | 10% | water-soluble P was reduced by 22.3% | [64] |
| wood ash | sandy loam | 7.4 | 10% | water-soluble P was reduced by 16.5% | [64] |
| wood ash | silt loam | 7.6 | | P concentration was reduced by 55.6% | [65] |
| sawdust | sandy soil | 7.8 | | P concentration was reduced by 58.1% | [65] |
| bauxite residues | | | 4% | the water-extractable P was reduced 95% | [66] |

### 3.2. Organic Amendment Materials

Compared with inorganic materials, organic materials have a positive effect on reducing the risk of P loss from soil to water [67]. The main organic amendment materials used are biochar and polyacrylamide (PAM) (Table 5).

Biochar is produced by biomass pyrolysis, and the raw materials used for biochar production include corn stover scale, rice straw, peanut husk, bamboo waste, bagasse, soybean straw, animal feces, etc. [68,69]. More recently, sewage sludge/biosolids biochar from nonindustrial treatment plants are being viewed as a way of recycling P and providing great P uptake efficiency. Biochar has a large specific surface area and abundant functional groups [70], which can change the cycle and availability of P by changing the adsorption and desorption performance of soil for P and can promote the adsorption and fixation of free $PO_4^{3-}$ in soil [71,72]. On the other hand, biochar can indirectly promote P retention by changing the structure of soil aggregates. Peng et al. [73] reported that the use of alkaline straw biochar in P-rich calcareous soil could effectively promote the conversion of unstable P to medium–high steady-state P, thereby reducing the availability of P. Xu et al. [74] reported that tree-derived biochar is conducive to the accumulation of thermally stable carbon, primarily aromatic carbon, which can provide additional adsorption sites for P and lower alkalinity, promoting the adsorption of P and inhibiting the effectiveness of P.

PAM is a type of polymer material with a crosslinked structure, which is polymerized from acrylamide and can promote the mutual condensation of fine soil particles to form

stable aggregates [75]. There are three main mechanisms for P retention from soil to water by PAM [76]: (1) through the interaction between soil particles and P to reduce the mobility of P in soil, which reduces the risk of P loss; (2) formed hydrogen bonds in soil solutions and high hydrophilicity that can reduce P loss from soil to water caused by soil erosion; and (3) through flocculation to convert soil colloidal P into soil particulate P, thereby reducing the migration of colloidal P from soil to water.

**Table 5.** Different organic amendment materials for P retention from soil to water.

| Organic Amendment Materials | Applied Soil Type | P Release Reduction | References |
|---|---|---|---|
| polyacrylamide | | the total P concentration of the leachate was decreased by 32.4% | [51] |
| anionic polyacrylamide | tea soil | total P reduced by 54% | [75] |
| maize stover biochar | corn-growing soil | inorganic P reduced by 3.3–59% | [77] |
| polyacrylamide modified biochar | paddy soil | total P reduced by 41.1% | [78] |
| Sugarcane-Derived Biochar | calcareous soil | / | [79] |
| wheat straw biochar | paddy-wheat rotation soil | the P utilization rate is increased by 38–230% | [80] |
| Rice-residue waste biochar | paddy soil | / | [81] |
| reed-biochar | paddy soil | total P reduced by 5.3–13.3% | [82] |
| maple and hickory sawmill waste biochar | forest soil | increases the absorption of a small amount of soluble P | [83] |

### 3.3. Composite Amendment Materials

Negatively charged materials and inherent low-adsorption capacity materials have limited P adsorption due to electrostatic repulsion and limited adsorption ability [57,84]. When using these materials, the addition of minerals or metallic or cationic surfactants to improve P retention efficiency is typically required [85,86]. Iron, aluminum, and magnesium oxides are common modified metal oxides, among which iron and aluminum are common metal oxides in soil with a large specific surface area and abundant functional groups. They show superior adsorption capacity for P by forming stable bonds with P(Fe-P, Al-P and P-O-Al/Fe). Metal oxides have been added to the surface of materials to fabricate composite amendment materials for reducing P loss from soil to water. Chen et al. [87] loaded magnesium onto cow manure biochar and found that magnesium-modified biochar could reduce the leaching amount of P by 89.25%, and the adsorption of P on the biochar surface also improved the oxidation resistance of biochar. Similarly, Zhao et al. [88] loaded the rare earth element lanthanum onto biochar and found that adding lanthanum-modified biochar could improve the adsorption capacity of soil for P and enhance the binding force between soil and P and that the adsorption capacity was stable and less affected by soil pH, which helps control the leaching of P from soil to water. Feng et al. [89] loaded cerium oxide onto corn stalk biochar (Ce-MSB) and found that adding Ce-MSB could reduce the TP concentration in surface water by 27.33%, which was 52.05% lower than that of MSB treatment.

### 4. Soil Retention Phosphorus Reloss to Water Environmental Risk

Soil properties, such as soil pH, soil inorganic and organic matter, and water condition, can cause the reloss process of soil P loss to the water environment.

### 4.1. Influence of Soil pH

P adsorbed in aluminum (hydrogen) oxides (Al-P) or calcium phosphate precipitates (Ca-P) is highly sensitive to pH changes. When the soil pH < 7, the protonated hydroxide radical generates positive charges on the surface of soil minerals, inducing the adsorption of negatively charged substances on the mineral surface through the formation of surface complexes, such as phosphate groups or phosphorus-containing organic molecules/colloids. When pH becomes neutral or slightly alkaline, soil particles become negatively charged, and the mineral surface repels negatively charged organic molecules/colloids containing P,

limiting the complexation of these P compounds on the soil mineral surface. Peng et al. [90] found that acidification tends to promote soil phosphorus loss into water bodies.

### 4.2. Influence of Soil Matter

Soil inorganic oxides have a certain effect on the adsorption capacity of soil for phosphorus. Mng'Ong'o et al. [91] reported that when the metal oxides content of aluminum, iron, and calcium were in the wide range of 234.56–3789.36 mg/kg, 456.78–2980.23 mg/kg, and 234.67–973.34 mg/kg, the adsorption capacity of the soil was higher than the average value under a high state, and the risk of phosphorus loss to water was lower. Therefore, some soils had deficient adsorption capacity, creating a risk of P loss to the water environment.

The soil organic matter is considered an important factor controlling the movement of P. An increase in organic matter provides a large amount of carbon and nutrients for microorganisms, promoting soil organic phosphate mineralization as the microbial population expands [92]. Organic acids produced by the decomposition of organic matter can decrease the precipitation of Ca-P minerals by increasing $H^+$, thereby increasing the concentration of soluble P in calcareous soil. Some of its active functional groups, such as the carboxyl group and phenolic hydroxyl group, can be complicated with metal ions (such as iron and aluminum) and reduce the available adsorption sites for P, promoting the release of soil phosphorus into the water.

### 4.3. Influence of Water Condition

Rainfall is the driving force of soil runoff and the main factor affecting phosphorus runoff into the water. Rainwater flows into farmland, and part of it is absorbed by crops and soil. When the moisture in soil reaches saturation, the excess rainwater gradually seeps down. With the continuous increase of rainfall, the loss of soil phosphorus to water increases. Yang et al. [93] reported that the runoff loss from soil P to water is not only related to rainfall but also closely related to rainfall intensity. When rainfall intensity increases, soil erosion will increase, and nutrients will be easily lost.

The flooding conditions will promote the reloss of P from soil to water. Under hydraulic erosion, soil phosphorus released by hydrolysis enters adjacent water along with surface runoff and soil erosion [90]. Xu et al. [94] demonstrated that residual P ($H_2SO_4$–$H_2O_2$ digestion) decreased by 18–27% in flooded paddy soil compared with aerobic soil. Zhang et al. [95] reported that in acidic soils, the effect of flooding conditions on P release was stronger, increasing by approximately 70%. Shaheen et al. [96] indicated flooding conditions increase soil pH and promote the dissolution of Fe and Al-P compounds in acidic soils, thereby increasing the risk of P loss.

### 5. Conclusions

With the rapid development of the economy, the application of P fertilizer has increased dramatically, and soil P nonpoint source pollution has caused serious water eutrophication. Currently, concerns over poor management of P nonpoint source pollution from soil to water and ongoing efforts to achieve environmental sustainability, especially the water environment, have piqued interest in soil restoration. Many studies have been conducted on the application of amendment materials to P nonpoint source pollution. This article reviews (1) different management strategies for P pollution from soil to water, including the knowledge of P forms and accumulation in soil, and P loss from soil to water bodies and management strategies; (2) the application of amendment materials to P nonpoint source pollution from soil to water, including recent advances in inorganic amendment materials, organic amendment materials, and composite amendment materials; and (3) the soil P retention reloss to water environmental risk. The theoretical investigations have provided insight into the research trends of soil P nonpoint source pollution to improve the understanding of using amendment materials to solve water pollution control.

## 6. Limitations of Existing Studies

The management of phosphorus pollution in water bodies has taken many results in the past, such as the removal of phosphorus from polluted water bodies by aquatic plants [97], the removal of phosphorus from polluted water bodies by amendment materials, and the modeling of the sources of total phosphorus pollution from nonpoint sources. However, due to the very complex composition of water bodies and the fluctuation of phosphorus concentration in water bodies due to climate and drastic human activities, phosphorus is still the primary form of pollution. There are still many problems to be improved in the management of phosphorus pollution in lakes and watersheds.

### 6.1. Watershed Phosphorus Pollution Management

Due to the complexity of the process of nonpoint source phosphorus pollution at the watershed scale, the many influencing factors and the late start of research on legacy effects, there are still many shortcomings and difficulties in process mechanisms and quantitative modeling.

In terms of process mechanisms, the understanding of phosphorus transport processes (especially subsurface runoff processes) in different hydrological pathways at the watershed scale is still relatively limited. Previous studies on nonpoint source phosphorus pollution processes in watersheds have focused on physical processes, such as soil erosion, with limited consideration of phosphorus biogeochemical processes (especially coupled with hydrological and biogeochemical regulation mechanisms). There is a lack of understanding of different media such as soil, groundwater, and sediment at the watershed or regional scale. The lack of research on phosphorus accumulation and its spatial distribution in different media such as soil, groundwater, and sediment at the watershed or regional scale has hindered the assessment and management of water environment pollution risks of phosphorus left in watersheds.

In terms of modeling studies, there is a lack of process models to describe the legacy effects of nonpoint source phosphorus pollution in watersheds. The understanding of the process mechanism of the legacy effects of nonpoint source phosphorus pollution is still unclear, and the current process models are still unable to comprehensively express the formation mechanism and action process of the legacy effects. The existing quantitative studies on the legacy effect of nonpoint source phosphorus pollution in watersheds are mainly on an interannual scale and basinwide analysis, and there is a lack of quantitative studies on the seasonal loss and spatial distribution of legacy phosphorus in different environmental media [98].

### 6.2. Management of Phosphorus Pollution in Lakes

At present, studies on lake phosphorus patterns are mainly focused on their important reservoir sediments or substrates, and on the differences between single lakes or different lakes in the same lake area; comparative studies on different types of lakes are relatively lacking.

Moreover, the management of eutrophication in phosphorus-controlled lakes remains a technical challenge, as there are no mature engineering techniques and experiences to draw on. Due to the influence of climate and intense human activities, phosphorus concentrations in lakes fluctuate greatly and even rebound in treatment, making it difficult to achieve phosphorus-control goals in lakes.

There are currently three forms of phosphorus in lake waters: water-soluble, particulate, and $PH_3$, of which the amount of $PH_3$ produced is small, and the relevant formation and transformation mechanisms are still unclear. The form of phosphorus in lake waters is more studied in the sediment-water transformation of inorganic phosphorus, while the transformation mechanisms of organic phosphorus between sediment and water bodies are yet to be further clarified. Most of the relevant lake water phosphorus management techniques are aimed at the reduction of total phosphorus concentrations, such as physical, chemical, and bioecological methods. The physical method, which has quicker results and

obvious restoration effects, is not long-lasting enough, and the research focuses on the efficiency of phosphorus removal by adsorbent materials, while less attention is paid to the mechanism of phosphorus adsorption. The chemical method, which is easy to operate but has high chemical costs and secondary ecological risks, can be used as an auxiliary or emergency control technology. The bioecological method is a comprehensive technology, which is the mainstream method of lake restoration at present, with low economic costs, good landscape effects, and stable restoration effects. Most of the relevant studies focus on the effect and mechanism of functional organisms (plants, animals, and microorganisms) and their grouping on the removal of total phosphorus, while the ecological safety of engineered bacteria and the risk of phosphorus removal mechanisms of aquatic animals still need to be further strengthened [99].

**Author Contributions:** X.G.: Conceptualization, methodology, validation, formal analysis, investigation, data curation, writing—original draft preparation; X.C.: Conceptualization, methodology, software, validation, investigation, data curation, writing—original draft preparation; M.L.: Conceptualization, methodology, validation, investigation, data curation, writing—original draft preparation; Y.Z.: writing—review and editing; C.W.: writing—review and editing; Y.W.: writing—review and editing; H.-T.T.: writing—review and editing; S.J.: writing—review and editing; T.Z.: Writing—review and editing, visualization, supervision, project administration, funding acquisition. All authors have read and agreed to the published version of the manuscript.

**Funding:** The research was sustained by the grant from the Undergraduate Research Program of China Agricultural University, the National Key Technology Research and Development Program of China [Grant number 2022YFE012907], the Government Purchase Service Project of Ministry of Agriculture and Rural Affairs of China [Grant number 202205510310600], the National Natural Science Foundation of China [Grant Number 31401944].

**Institutional Review Board Statement:** Not applicable.

**Data Availability Statement:** Not applicable.

**Conflicts of Interest:** The authors declare no conflict of interest.

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
