# Peer review of "Toward a Better Understanding of Phosphorus Nonpoint Source Pollution from Soil to Water and the Application of Amendment Materials: Research Trends"

_water, doi:10.3390/w15081531_

Round 1
Reviewer 1 Report
Revisione water 6 marzo 2023
The Authors in the review submitted to Water address a very important issue, eutrophication due to the dispersion of phosphorus present in fertilizers in water. This represents a serious environmental problem widespread in the world and it is urgently necessary to remedy it. The Authors introduce the topic well, framing the question well with adequate bibliographic references. They are equally comprehensive when considering possible amendments to prevent P pollution. The authors also indicate that studies will be needed to evaluate the effects, including collateral effects, of the amendments, but already in this review they could critically analyze the possible economic and environmental consequences of the countermeasures examined. For example, rare earth elements are expensive, aluminum can be released into soil at acidic pH, and aluminum toxicity is now much investigated. polyacrylamide, a non-toxic polymer of toxic acrylamide monomers, can still release its constituents with serious consequences.
Margareta Törnqvist et al., 2000 - Leakage of acrylamides from a tunnel construction work: exposure monitoring and health effects to humans and animals (https://www.eu-alara.net/images/stories/pdf/program4/An-Tornqvist.pdf)
Scientific Opinion on acrylamide in food1 EFSA Panel on Contaminants in the Food Chain (CONTAM)2, 3 European Food Safety Authority (EFSA), Parma, Italy (https://www.efsa.europa.eu/it/efsajournal/pub/4104).
The authors could mention the possible consequences on the microbial communities of the soil and the composition of humic acids, balances that must be strictly controlled and maintained. In practice, a hint could be made whether even the proposed countermeasures would lead to serious environmental problems or would not be economically sustainable
Some notes for the Authors
Please beware of repetitions that weigh down the text, pay attention to the spaces that are sometimes missing, especially between the text and the parentheses of the references
Please rewrite line112-116, now is difficult to read: China is a large agricultural country, the amount of fertilizer application has reached 100 million t, the level of fertilizer application is 2.6 times higher than the world level of fertilizer application, while the utilization rate of fertilizer application is only 30-50% of the level, a large number of P components into water bodies through various ways.
Please citation not in this way [34], [36], but [34,36]
Please rewrite lines 311-315, speeches appear a bit disconnected: For example, the alumina could be extracted from alum sludge using different chemical treatment methods after being subjected to different thermal treatments[51]. Red mud is a byproduct of the Bayer process to extract alumina from bauxite[52]. Wheat straw biochar and corn straw biochar can be produced from wheat rice and corn straw, respectively. Fly ash is solid waste after the combustion of pulverized coal in coal-fired power plants [53]
Please attention to line 432: They show superior adsorption capacity for P by forming stable bonds (Fe-P, Al-P 和 P-O-Al/Fe).
Please pay attention to two-line sentences in the tables. If this is not due to the building of the manuscript, the formatting of the tables must be changed
The Authors have worked with great dedication and seriousness and their review is noteworthy, but for publication it needs improvement. In my opinion minor/major revision
Author Response
Response to Reviewer 1
Thank you for your constructive and encouraging comments on our paper. We have learned much from your comments. According to your comments, we have revised our paper point by point and response the comments carefully. The manuscript is revised and marked with new line numbers and the lists of major changes are as follows.
- Please beware of repetitions that weigh down the text, pay attention to the spaces that are sometimes missing, especially between the text and the parentheses of the references
Thank you so much for your comments! We've corrected the duplicate words and filled in the missing spaces in the paper.
- Please rewrite line112-116, now is difficult to read: China is a large agricultural country, the amount of fertilizer application has reached 100 million t, the level of fertilizer application is 2.6 times higher than the world level of fertilizer application, while the utilization rate of fertilizer application is only 30-50% of the level, a large number of P components into water bodies through various ways.
Thank you so much for your comments! We've rewritten line112-116 into “China is a large agricultural country with a fertilization amount of 100 million tons. The average fertilization amount is 2.6 times the world average, but the fertilization utilization rate is only 30-50% of the world average. A great amount of phosphorus enters the water body through various channels.” in line 111-114.
- Please citation not in this way [34], [36], but [34,36]
Thank you so much for your comments! We've changed the citation to [34,36] in Line 202.
- Please rewrite lines 311-315, speeches appear a bit disconnected: For example, the alumina could be extracted from alum sludge using different chemical treatment methods after being subjected to different thermal treatments[51]. Red mud is a byproduct of the Bayer process to extract alumina from bauxite[52]. Wheat straw biochar and corn straw biochar can be produced from wheat rice and corn straw, respectively. Fly ash is solid waste after the combustion of pulverized coal in coal-fired power plants [53]
Thank you so much for your comments! We've rewritten the sentences in line 312-327.
- Please attention to line 432: They show superior adsorption capacity for P by forming stable bonds (Fe-P, Al-P 和 P-O-Al/Fe).
Thank you so much for your comments! We've changed “They show superior adsorption capacity for P by forming stable bonds (Fe-P, Al-P å’Œ P-O-Al/Fe).” Into “They show superior adsorption capacity for P by forming stable bonds with P(Fe-P, Al-P å’Œ P-O-Al/Fe)” in line 457-458.
- Please pay attention to two-line sentences in the tables. If this is not due to the building of the manuscript, the formatting of the tables must be changed.
Thank you so much for your comments! We have adjusted the layout of the table in the paper.
Reviewer 2 Report
This paper has presented some valuable information of the efficacies of amendment materials on the reduction of P release from soil to water. Its relevancy for the targeted special issue (Sustainable Wastewater Treatment and the Circular economy) is not clear. The paper does not have a good structure and its English writing need significant improvement. It is difficult to comprehend in some parts. Some key information is missing. Some of the main issues are listed below for authors to address:
1) Introduction: the introduction is too long and lack of focus on the topic to be considered. Some brief description about the P loss from soil and its effects on aquatic ecology, e.g. eutrophication is expected but I do not see how the source apportionment and spatial patterns are useful here. Please consider whether Figure 2/3/4/5 are all really necessary. Section 1 need restructuring to make it concise, logical and relevant to the topic for the special issue.
2) Methodology on literature retrieval. Please provide details on how the literature were collected, e.g. keywords used, time period covered and database searched. The authors have used the 'non-point source pollution'. Another widely used term is 'diffuse pollution'. Will the figure 5 change with the use of a different search term? The Figure 6 is innovative and should be kept.
3) Result presentations: Authors have shown the soil types for the selected studies. Please give relevant soil classification systems used. If possible, please use one system for all the studies. 'Sandy soil' is too general a term. Other relevant factors could also be shown if available, such as organic matter content, annual rainfall, land use, etc. Please comment on how widely these materials are being used at moment and where.
4) Limitations of existing studies: Please discuss what are the main limitations / shortcoming of existing studies. What are the main challenges and research areas in the near future. As a paper on research trend, this should be covered.
5) I have attached an annotated version of the paper where more detailed questions are raised which need authors' attention.

Author Response
Response to Reviewer 2
Thank you for your constructive and encouraging comments on our paper. We have learned much from your comments. According to your comments, we have revised our paper point by point and response the comments carefully. The manuscript is revised and marked with new line numbers and the lists of major changes are as follows.
- Introduction: the introduction is too long and lack of focus on the topic to be considered. Some brief description about the P loss from soil and its effects on aquatic ecology, e.g. eutrophication is expected but I do not see how the source apportionment and spatial patterns are useful here. Please consider whether Figure 2/3/4/5 are all really necessary. Section 1 need restructuring to make it concise, logical and relevant to the topic for the special issue.
Thank you so much for your comments! We have restructured section 1 to make it concise, logical and relevant to the topic for the special issue in section 1.
- Methodology on literature retrieval. Please provide details on how the literature were collected, e.g. keywords used, time period covered and database searched. The authors have used the 'non-point source pollution'. Another widely used term is 'diffuse pollution'. Will the figure 5 change with the use of a different search term? The Figure 6 is innovative and should be kept.
Thank you so much for your comments! We have explained in detail how the literature were collected in line184-187.
- Result presentations: Authors have shown the soil types for the selected studies. Please give relevant soil classification systems used. If possible, please use one system for all the studies. 'Sandy soil' is too general a term. Other relevant factors could also be shown if available, such as organic matter content, annual rainfall, land use, etc. Please comment on how widely these materials are being used at moment and where.
Thank you so much for your comments! We have reclassified the soil according to the classification and codes for Chinese soil.
Some research documents only provide a simple description of the selected research soil and cannot be accurately classified based on the classification system.
In addition, many current studies on the use of improved materials for soil phosphorus fixation are limited to laboratory research and some short-term experiments, and may not have specific locations and scope of use.
- Limitations of existing studies: Please discuss what are the main limitations / shortcoming of existing studies. What are the main challenges and research areas in the near future. As a paper on research trend, this should be covered.
Thank you so much for your comments! We have filled in the main limitations in section 6.
- I have attached an annotated version of the paper where more detailed questions are raised which need authors' attention.
Thank you so much for your comments! We have corrected the paper according to the annotated version.